# Healthcare provider perspective on barriers and facilitators in the care of pediatric injury patients at a tertiary hospital in Northern Tanzania: A qualitative study

**Elizabeth M. Keating**[1]*, **Francis Sakita**[2,3], **Kajsa Vlasic**[1], **Ismail Amiri**[2], **Getrude Nkini**[2], **Mugisha Nkoronko**[2,3], **Bryan Young**[1], **Jenna Birchall**[1], **Melissa H. Watt**[4], **Catherine A. Staton**[5,6,7], **Blandina T. Mmbaga**[2,3,7,8]

**1** Division of Pediatric Emergency Medicine, Department of Pediatrics, University of Utah, Salt Lake City, Utah, United States of America, **2** Kilimanjaro Christian Medical Centre, Moshi, Tanzania, **3** Kilimanjaro Christian Medical University College, Moshi, Tanzania, **4** Department of Population Health Sciences, University of Utah, Salt Lake City, Utah, United States of America, **5** Department of Emergency Medicine, Duke University Medical Center, Durham, North Carolina, United States of America, **6** Global Emergency Medicine Innovation and Implementation (GEMINI) Research Center, Duke University Medical Center, Durham, North Carolina, United States of America, **7** Duke Global Health Institute, Duke University, Durham, North Carolina, United States of America, **8** Kilimanjaro Clinical Research Institute, Moshi, Tanzania

* Elizabeth.Keating@hsc.utah.edu

**Data Availability Statement:** The data for this manuscript is covered under a data sharing agreement, and thus we are unable to openly share

## Abstract

Pediatric injuries are a leading cause of morbidity and mortality in low- and middle-income countries (LMICs). The recovery of injured children in LMICs is often impeded by barriers in accessing and receiving timely and quality care at healthcare facilities. The purpose of this study was to identify the barriers and the facilitators in pediatric injury care at Kilimanjaro Christian Medical Center (KCMC), a tertiary zonal referral hospital in Northern Tanzania. In this study, focus group discussions (FGDs) were conducted by trained interviewers who were fluent in English and Swahili in order to examine the barriers and facilitators in pediatric injury care. Five FGDs were completed from February 2021 to July 2021. Participants (n = 30) were healthcare providers from the emergency department, burn ward, surgical ward, and pediatric ward. De-identified transcripts were analyzed with team-based, applied thematic analysis using qualitative memo writing and consensus discussions. Our study found barriers that impeded pediatric injury care were: lack of pediatric-specific injury training and care guidelines, lack of appropriate pediatric-specific equipment, staffing shortages, lack of specialist care, and complexity of cases due to pre-hospital delays in patients presenting for care due to cultural and financial barriers. Facilitators that improved pediatric injury care were: team cooperation and commitment, strong priority and triage processes, benefits of a tertiary care facility, and flexibility of healthcare providers to provide specialized care if needed. The data highlights barriers and facilitators that could inform interventions to improve the care of pediatric injury patients in Northern Tanzania such as: increasing specialized provider training in pediatric injury management, the development of pediatric injury care guidelines, and improving access to pediatric-specific technologies and equipment.

these data. For data access requests, please contact our non-author KCMC representative Gwamaka William at gwamakawilliam14@gmail.com.

**Funding:** EMK was supported in this work by the Fogarty International Center of the National Institutes of Health (D43 TW009337) and the Eunice Kennedy Shriver National Institute of Child Health and Human Development (1K23HD112548-01). The content is solely the responsibility of the authors and does not necessarily represent the official views of the National Institutes of Health. The funders had no role in study design, data collection and analysis, decision to publish, or preparation of the manuscript.

**Competing interests:** The authors have declared that no competing interests exist.

## Introduction

More than 5 million individuals die annually of injuries [1] and studies have shown that over 95% of fatal injuries occur in low- and middle-income countries (LMICs) [2–5]. While injuries affect all age groups, increasing rates of injury affecting children globally are of particular public health concern [6]. The child injury death rate is 3.4 times higher in LMICs than in high-income countries [7], and pediatric injuries are a leading cause of morbidity and mortality in LMICs. While significant gains have been made in reducing child mortality toward the Sustainable Development Goals (SDGs) [8], this progress has focused primarily on infectious diseases and neglected the burden of injuries among children [9, 10]. In order to close the gap on childhood survival, more emphasis needs to be placed on the growing problem of pediatric injury.

While adult trauma registries describing injury rates in LMICs are more commonplace, pediatric trauma registries in LMICs are not as well documented [11]. To date, pediatric injury registries in LMICs have shown mortality rates of injured children ranging from 0.3% to 8.2% [11–16].

The recovery of injured children in LMICs is often impeded by barriers in accessing and receiving timely and quality care at healthcare facilities. Healthcare providers in LMICs often face a variety of barriers when treating patients, such as resource limitations, inadequate staffing, large patient loads, lack of pre-hospital and transport services, financial difficulties for families, delayed presentation for illnesses and injuries, cultural differences, and a lack of specialist care [9, 17, 18]. Due to these challenges, it is important to investigate how these barriers impede quality of care for injured children.

There is a general lack of data describing barriers and facilitators in pediatric injury care from the perspective of healthcare providers in LMICs. It has been estimated that high-quality hospital care would decrease child injury mortality rates by eight percent [19]. Thus, the healthcare provider perspective on experiences caring for pediatric injury patients is an important component in combination with pediatric injury epidemiology in LMICs to ultimately provide appropriately targeted high-quality hospital care interventions aimed at decreasing pediatric injury morbidity and mortality. Although there have been some qualitative studies evaluating provider perspectives in global trauma care, most of this data describes adult emergency care systems [17, 18, 20]. Thus, there is a knowledge gap in barriers and facilitators from healthcare providers who directly care for and provide services to injured children in LMICs.

The objective of this study was to identify the barriers and facilitators in pediatric injury care from the perspective of health care providers at Kilimanjaro Christian Medical Center (KCMC), a tertiary zonal referral hospital in Northern Tanzania. This data can be used to improve health systems to support injured pediatric patients at KCMC and in similar tertiary care settings in LMICs.

## Methods

### Overview

This is a qualitative study involving semi-structured focus-group discussions (FGDs). The Standards for Reporting Qualitative Research checklist was used and is included in S1 Checklist. In order to examine the barriers and successes in pediatric injury care, grounded theory methodology was utilized with an interpretivist paradigm. This study and all procedures received ethical approval from the institutional review boards at the Tanzanian National Institute for Medical Research (NIMR/HQ/R.8a/Vol.IX/3475), Kilimanjaro Christian Medical Centre (1252), and the University of Utah (IRB_00134560). Written consent was obtained from all participants.

This study was performed adjacent to a previously published analysis which described epidemiologic findings from the first pediatric injury registry in Northern Tanzania [11].

## Inclusivity in global research

Additional information regarding the ethical, cultural, and scientific considerations specific to inclusivity in global research is included in S2 Checklist.

## Study setting

This study was conducted at Kilimanjaro Christian Medical Centre (KCMC), a tertiary zonal referral hospital for Northern Tanzania (including regions of Arusha, Manyara, Kilimanjaro, and Tanga) with a catchment population of 12 million. It is one of four tertiary zonal referral hospitals in Tanzania and serves as a referral site for children with injuries requiring specialty investigations, imaging, and treatment. Injured children are most often seen first at the Emergency Medical Department (EMD) at KCMC, which sees approximately 1400–1700 pediatric patients per year. Injured children who require inpatient care are then admitted mainly to one of the following wards: Pediatric, Surgical, or Burn. KCMC employs pediatricians that often assist in the care of pediatric trauma patients. In addition, KCMC employs general surgeons and orthopedic surgeons who treat injured children. Consultations are also provided within the departments. There is also a specialized Burn Ward at KCMC. Each of the wards above has nurses that work exclusively in these wards.

## Participants

Healthcare providers including physicians and nurses were eligible for semi-structured FGDs if they cared for pediatric injured patients in the EMD or during their hospitalization at KCMC, were fluent in Kiswahili or English, and could provide informed consent. In order to adequately represent the healthcare provider population, we attempted to include both doctors and nurses from each of the wards that cares for pediatric injury patients. A research assistant contacted doctor or nursing leads for each ward and informed them about the focus group discussion. Each lead contacted healthcare providers in their ward either in person or via telephone to invite them to participate. A convenience sample of 30 healthcare providers was recruited. The number of participants recruited was pre-set and informed by team-based discussions of when thematic saturation was likely to occur.

## Qualitative procedures

Five FGDs were completed with 30 healthcare providers from the EMD, Burn Ward, Surgical Ward, and Pediatric Ward that care for injured patients at KCMC. There were 14 (47%) doctors and 16 (53%) nurses that participated in the FGDs. The discussion guide was developed in order to identify barriers and facilitators to providing high quality and efficient care to pediatric injury patients. Participants were asked about: training/preparation, resources, and quality of care that injured children receive in the emergency department; the transition from the emergency department to the ward; training/preparation, resources, and quality of care that injured children receive after admission in the ward; and the discharge process. The guide was translated from English to Kiswahili by two native Kiswahili speakers, checked for accuracy in translation meetings by a team of native speakers, and pilot tested by the research team.

FGDs were conducted between February 2021 and July 2021. All FGDs were completed in Kiswahili by two trained bilingual Tanzanian research assistants (authors IA and GN) who had extensive experience conducting qualitative discussions. FGDs were conducted in quiet, private meeting rooms within KCMC in each of the wards. Each FGD took approximately one

to two hours. Participants were provided a copy of the consent form, which was read aloud by research assistants. Participants provided their written informed consent prior to completing the FGDs and received refreshments during the discussion. FGDs were audio recorded for verbatim transcription, and the interviewers took notes during the discussion to assist with transcription. FGD audio recordings were transcribed and translated to English by the two bilingual Tanzanian research assistants.

### Research team and reflexivity

Focus groups were conducted by Tanzanian research assistants GN (female) and IS (male), who are fluent in English and Kiswahili, experienced in qualitative interviewing, and were supervised by the PI. The research assistants completed two weeks of training that included qualitative FGD strategies and gaining familiarity with the study protocol and qualitative discussion guides. There was no relationship between the interviewer and the participants prior to the focus groups. The participants were informed about the goals of the interview and reasons for doing the research prior to the focus groups. There was no interaction between the participants and the rest of the authorship team, however the authorship team did interact with the data during analysis and manuscript preparation. Data analysis was conducted by three American researchers (two female and one male). Preliminary findings were presented and discussed with the multi-national team that included five Tanzanians (two female and three male) and six Americans (five female and one male) who brought expertise in global injuries, qualitative research, and global child health research.

### Qualitative data analysis

De-identified transcripts of the FGDs were analyzed using a team-based, thematic analysis approach [21]. Analysis relied heavily on memo writing as a strategy to synthesize data, make connections and point out contradictions, and identify patterns both within and across the data sources [22]. After multiple readings of the FGD transcripts, a memo summarizing each transcript was written by investigator BY after training by EMK and MHW. Memos followed an established template of a priori domains, informed by the interview guide, to extract and synthesize the text's core meaning related to the research questions and to extract representative quotes. Each memo was on average seven single-spaced pages long and reviewed by EMK to ensure that the interview content was being interpreted appropriately. After the memos were reviewed, investigators BY, EMK, and MHW met to reach team consensus on the emerging themes and finalize the codebook. The memos were then coded in Dedoose software, and the codebook was continually adapted to reflect new or emerging themes as coding progressed. All memos were double-coded by BY and JB, with intercoder discrepancy discussed amongst the research team and resolved by consensus, with EMK determining final application of codes if disagreements remained. Authors did not have access to information that could identify individual participants during or after data collection.

## Results

Focus group discussions (FGDs) revealed themes related to both barriers and facilitators (Table 1) of high-quality pediatric injury care.

### Barriers to quality pediatric injury care

Five themes emerged regarding barriers to quality pediatric injury care: 1) lack of pediatric-specific injury training and care process guidelines, 2) lack of appropriate medical equipment,

**Table 1. Themes related to barriers and facilitators in providing quality pediatric injury care.**

|  | *Emerging Themes* |
|---|---|
| *Barriers* | Lack of pediatric-specific injury training and care guidelines |
|  | Lack of appropriate pediatric-specific equipment |
|  | Staffing shortages |
|  | Lack of specialist care |
|  | Complexity of cases due to cultural and financial challenges |
| *Facilitators* | Team cooperation and commitment |
|  | Strong priority and triage processes |
|  | Benefits of a tertiary care facility |
|  | Flexibility of healthcare providers to provide specialized care if needed |

3) shortage of staffing, 4) lack of specialist care, and 5) complexity of cases due to pre-hospital delays in patients presenting for care caused by cultural and financial barriers.

**Lack of pediatric-specific injury training and care guidelines.** Healthcare providers reported that they lack pediatric-specific emergency and injury training and that there is a general lack of pediatric-specific injury care guidelines at KCMC. As a result, when pediatric patients with injuries present to KCMC, they often do not have institutional processes to help them navigate these pediatric-specific medical needs.

*"We aren't really trained to handle this situation. Especially for pediatric emergencies. We got training under MD but not specifically for emergencies. . .it's more about when you see it, you try to do better next time. Maybe lack of training. Specifically, children and also in general."* (EMD)

*"There are guidelines for emergencies, but not specific for pediatric emergencies. Having guidelines will help medical staff act quicker with patients."* (EMD)

*"I think those guidelines should be specifically for children because. . .when pediatric trauma happens, we usually follow the general ward guidelines simply because it is trauma, but we don't have specifically for pediatric [trauma] that we have to do this and this."* (EMD)

Many described relying on their anecdotal experiences caring for injured children and wishing they had more training and experience with pediatric care.

*"We save them through experience and asking [our] fellows who are familiar on how to save these children."* (Pediatric Ward)

*"[We] need more training in pediatric trauma."* (Surgical Ward)

**Lack of appropriate medical equipment.** Providers reported that there is a general lack of pediatric-specific equipment in the EMD needed for injured patients. As a referral hospital for injured children, they do not reliably have pediatric-sized instruments, which directly affects the care healthcare providers are able to provide. They also described lack of transportable oxygen, limited number of pediatric-specific beds, and delayed medication availability affecting patient care.

*"Generally, we don't have enough equipment to save children. For example, if you want to intubate a child those laryngoscopes are not fitting the child. I can say there is not much equipment for children."* (EMD)

*"Shortage of [pediatric-specific] beds causes delays in admission; sometimes [we] must look to other wards to admit patients, or patients must wait in EMD until a bed is ready." (Pediatric Ward)*

*"Not enough medicine to supply injured children." (Surgical Ward)*

**Staffing shortages.** Healthcare providers described difficulty staffing shifts in the EMD. The inconsistent coverage is difficult especially when the EMD is busy. The ward providers also noted a shortage of doctors, and made suggestions for potential staffing changes.

*"In short, we don't have enough staff for all shifts, even morning shifts. Evening shifts we have shortage of staff and days are unpredictable at EMD. There are days when there are few patients and other days a lot of patients, so staff are not enough." (EMD)*

*"[I] suggest since we have shortage of doctors. . .they could help us arrange the shift of intern doctors [so that] one doctor will remain in the ward and another one will be working in the-ater." (Surgical Ward)*

**Availability of specialist care.** Providers described availability of specialist care as a bar-rier for injured pediatric patients. Pediatric-specific subspecialist care, such as pediatric neuro-surgeons, pediatric orthopedic surgeons, and pediatric surgeons, are not consistently available at KCMC. The neurosurgeons, orthopedic surgeons, and general surgeons that are able to care for these patients are often very busy given the large volume of cases in both adults and pediat-rics. Due to the limited availability of specialists, often there is a delay in admitting injured children from the EMD while waiting for specialist evaluation. For those patients who are admitted, limited availability of specialist providers leads to generalist providers making care decisions for injured patients without appropriate guidance.

*"Another thing we depend on. . .special to trauma cases [is] where the child should be admitted. . . This also sometimes brings the delay for the admission as we wait long at the emergency for the review of other specialists [t]o come and make a decision." (Pediatric Ward)*

*"[There are not enough] specialists in the wards and in most cases the specialist are [needed for] severe cases of the patients by the residents. [They are also] sometimes not available so. . .the intern tries to solve the cases alone." (Surgical Ward)*

*"[We] mostly have general specialists available; [we] lack specialists for pediatric patients." (Surgical Ward)*

**The continuum of delays to in-hospital care caused by cultural and financial barriers.** FGDs also provided perspective from healthcare providers on cultural and financial barriers that cause both pre-hospital and in-hospital delays and increase the complexity of cases. There was general recognition that many families have strong beliefs in traditional medicine and will seek out these medicinal practices outside of the hospital first which often leads to delays in care.

*"[An] additional point is that many parents have traditions and values, so when their child gets burn injuries they believe that when they give certain kind of medicine or they do certain rituals this child will recover. So they have their own medicine [in] every area which they [believe] will help to cure [the] child's wound." (Burn Ward)*

In addition, healthcare providers reported that financial barriers can delay access to care for pediatric injury patients after arrival to the hospital. The unanticipated medical costs associated with injury can lead to delays in care as some of these costs need to be paid for at the time of treatment. Needs like medications, procedures, and hospitalization all require payment in cash around the time of receipt when the patient does not have health insurance. Some patients require exemption from the Social Welfare Officers to receive timely treatment if cost is a barrier.

"*The patient may come to the ward but still the issue of payment will delay everything. The patient is to go to the operation room but has no money. The social [welfare officers] will have to come and exempt the child [before] he can be operated on.*" *(Surgical Ward)*

"*Medication sometimes is a challenge as patients have to buy them alone. So, sometimes you may give the relatives a prescription of the medicine and they fail to buy until they communicate with other family members to get the money to buy, hence the patient may delay getting medication.*" *(Surgical Ward)*

**Facilitators to quality pediatric injury care.** Four themes emerged regarding facilitators to quality pediatric injury care: 1) team cooperation and commitment, 2) prioritization and triage of pediatric injury patients, 3) benefits of a tertiary care facility, and 4) flexibility of healthcare providers to provide specialized care if needed.

**Team cooperation and commitment.** Healthcare providers largely agreed that when a pediatric injury patient is seen in the EMD or is admitted for care in the hospital there is efficient team effort aimed at providing appropriate care despite the barriers described above.

"*Team cooperation; Patients are received quickly; all competent staff prioritize pediatric emergencies so they get the best care as fast as possible.*" *(EMD)*

"*Healthcare providers are passionate about their work and the care they provide to patients.*" *(Burn Ward)*

**Priority and triage.** Upon initial presentation to the EMD, providers reported pediatric injury patients receiving prioritized care. Effort is particularly aimed at those with more critical injuries or clinical instability.

"*Patients are received quickly; all competent staff prioritize pediatric emergencies so they get the best care as fast as possible.*" *(EMD)*

"*[Healthcare providers] have a strong system of prioritizing more critical patients to speed up transfer to [the] specific ward.*" *(EMD)*

**Benefits of a tertiary care facility.** Compared to other health facilities in Northern Tanzania, there are benefits to KCMC as a tertiary care facility. These include more pediatric-specific medical equipment available than other facilities in the area, and more human resources such as specialists. Providers recognized that the increased availability of resources and equipment for pediatric injury patients treated in the EMD and the wards is beneficial for their outcomes.

"*More pediatric-specific equipment [present than other facilities.]*" *(EMD)*

"*We have resources at KCMC that other hospitals do not have.*" *(Surgical Ward)*

**Flexibility of healthcare providers to provide specialized care if needed.** Providers reported that many pediatric injury patients require care by specialist providers in the EMD or once admitted to the hospital. When there are delays in specialist availability for these patients, the healthcare providers at KCMC are flexible and provide the best care they can in order to prevent complications.

"*Investigations take place by other departments. . .the [patients] are being seen by certain specialists and plans of care are being discussed.*" (Surgical Ward)

"*Health providers try their best, even if there are delays from surgical doctors seeing patients.*" (Pediatric Ward)

## Discussion

This is a study of barriers and facilitators in the care of pediatric injury patients from the healthcare provider perspective at a tertiary referral hospital in Northern Tanzania. Healthcare providers faced significant systems-level barriers to providing quality care for children presenting with injuries, but they also expressed great resilience, flexibility, and commitment to meeting the needs of these injured children. In order to provide quality care for injured children at KCMC, they need resources including specialized training in caring for injured children, the development of pediatric injury care guidelines at the systems level, and increased availability of appropriate pediatric-specific equipment. These barriers and facilitators not only provide insight on the care of pediatric injured patients that can lead to interventions at KCMC to improve care and outcomes, but also could be translated to other tertiary referral hospitals in LMIC settings.

A barrier discussed often by the healthcare providers in our study was the lack of pediatric-specific injury training. This is a known challenge in LMICs, and specialists trained in the care of pediatric trauma patients are in short supply in Tanzania [23]. Few hospitals in LMICs have emergency medicine-trained specialists, and even fewer have pediatric emergency medicine specialists that have unique training in caring for acutely ill and seriously injured children. This leads to the majority of pediatric injury care being provided by healthcare providers with no specialized training in the management of pediatric trauma [9]. The lack of trained providers has been shown to have a significantly negative impact on outcomes [24]. To combat this, more subspecialist training programs are needed in sub-Saharan Africa. Ideally, there should be a subspecialty training in pediatric emergency medicine. In addition, pediatric training is needed in the surgical specialties, as well, such as neurosurgery, orthopedic surgery, and general surgery. There are rarely providers in LMICs with pediatric-specific training in these specialties currently. Thus, a potentially more achievable solution would be to incorporate trauma management training into the medical school curriculum in LMICs in order to ensure widespread dissemination of the skills needed to manage pediatric trauma [9]. In the absence of training specialists, opportunities to improve the pediatric-specific skills of providers are needed via short courses or trainings. Specific trainings on caring for pediatric trauma patients in LMICs do exist, and one example is the Emergency Triage, Assessment, and Treatment plus (ETAT+) training that has resulted in better prioritization of pediatric trauma cases in the ED [25]. Telesimulation is another option for teaching pediatric trauma skills to healthcare providers in LMICs [26]. An excellent resource for online learning and simulation is OPEN Pediatrics, an online community of clinicians that share best practices from around the world [27].

In high-income countries, health systems interventions such as care process guidelines have been shown to standardize care and improve outcomes in pediatric injuries. These care process guidelines have not been widely developed for pediatric injured patients in LMICs [28–

30]. One review found only one pediatric traumatic brain injury guideline developed in an LMIC, with guidelines for other types of injuries similarly rare [31]. In the adult trauma population, however, some institutions in LMICs have developed protocols and guidelines for trauma management that have led to improved outcomes [26, 32–35]. Organizations such as the World Health Organization and African Federation of Emergency Medicine have provided guidelines on the appropriate resources needed for the care of pediatric patients which can be adjusted depending on the availability of local resources [26, 33–36]. At KCMC, as we saw in our study, care process guidelines for injured children have not been adopted and care has not been standardized [37]. Thus, there is an opportunity to derive locally-relevant standardized protocols in order to increase use of timely appropriate interventions for pediatric trauma patients and decrease mortality rates [38].

Our study findings highlighted the shortages of staff and specialists in caring for pediatric injured patients. These shortages are common in LMICs including Tanzania, and many studies have described inadequate staffing levels as barriers leading to delays in assessment in treatment [39]. Regarding the lack of specialists such as surgical services, in HICs surgical teams form a trauma management team that facilitates timely surgical intervention when needed and has shown improved outcomes [9]. KCMC has a strong general surgery team, but given high patient volumes they are often overworked and unavailable. Further, they have only one general surgeon with specialized pediatric training, and this one surgeon is unable to cover all pediatric injury patients. Thus, a pediatric surgeon-led trauma team and related resources are not currently feasible in most low-resource areas of the world including KCMC, but remain a goal for the future [9].

Healthcare providers in our study noted the shortages of pediatric-specific equipment. This is important to note as injured children require special considerations including different equipment and approaches to airway stabilization, and reducing diagnostic radiation exposure with imaging [28]. Caring for injured children requires adequate smaller sized equipment for their acute stabilization. There are significant differences even within the same LMIC with regard to available resources for emergency room care between different levels of healthcare facilities [9]. We saw this in our study when the healthcare providers reflected on one of the facilitators which was having relatively more resources at KCMC than other hospitals in the region. It would be helpful to increase the availability of pediatric-specific trauma and resuscitation equipment.

Another theme noted in our study was increased complexity of cases due to both pre-hospital and in-hospital delays in patients presenting for care caused by cultural and financial barriers. One such delay was the use of traditional medicine prior to seeking hospital care. Use of traditional medicine administered by traditional health practitioners is common in LMICs, and one study showed that patients who used these methods presented later and had higher mortality [40–42]. Our group recently published a paper reporting delays to care from the perspective of pediatric injury patients' caregivers [43]. Our findings from the healthcare providers' perspective are the same regarding the presence of cultural and financial barriers often causing delays. Our findings are further supported by a study on barriers to pre-hospital emergency care in an LMIC that reported cultural beliefs are often a barrier in the decision to seek medical attention [17]. Other studies have shown that often more familiar traditional approaches are preferred to newer approaches when it comes to emergency care [44–47].

## Limitations

One potential limitation of our study is social desirability bias, which could have made healthcare providers less comfortable in discussing barriers to pediatric injury care at KCMC.

However, we discovered as many barriers as facilitators, so we predict that this bias did not significantly limit our findings. Similar to social desirability bias, an additional limitation is that we did not provide an opportunity for participants to provide anonymous responses since data collection was in focus group discussions only. Further studies on barriers and facilitators should provide this opportunity to get more individualized responses. Another limitation is that we were only able to conduct a limited number of FGDs with 30 healthcare providers, so it is possible that there are views we missed from other providers. However, we feel that a sample size of 30 provided a robust example of the opinions of the majority of the healthcare providers at KCMC, and thematic saturation was reached. We also included both doctors and nurses to allow for varied perspectives. While a benefit, including nurses and doctors in the FGDs can also be viewed as a limitation. Previous qualitative studies in LMICs have shown inter-professional rivalry to be a theme that negatively impacts patient care [48–50], but did not present during our study. This theme may not have presented itself because the mixed presence of nurses and doctors may have led participants to not feel free to discuss this theme. A final limitation is the fact that this study was conducted at a tertiary zonal referral hospital, which is not reflective of care more broadly in Tanzania at other levels of care, such as dispensaries, health centers, district hospitals, or regional hospitals. Thus, the conclusions made in this study may only be applicable to tertiary hospitals in similar settings.

## Conclusions

In this study of healthcare providers at a tertiary referral hospital in Northern Tanzania, we identified both barriers and facilitators to the care of pediatric injury patients. Healthcare providers faced significant systems-level barriers to providing quality care for children presenting with injuries, but they also expressed great resilience, flexibility, and commitment to meeting the needs of these injured children. In order to provide quality care, healthcare providers need resources including specialized training in caring for injured children, the development of pediatric injury care guidelines at the systems level, and increased availability of appropriate pediatric-specific equipment. These findings could lead to interventions to improve the care of pediatric injury patients at KCMC in Northern Tanzania that could be translated to other tertiary referral hospitals in LMIC settings.

## Supporting information

**S1 Checklist. Standards for reporting qualitative research.**
(PDF)

**S2 Checklist. Inclusivity in global research.**
(DOCX)

## Acknowledgments

The authors would like to acknowledge the healthcare providers at KCMC who participated in our study.

## Author Contributions

**Conceptualization:** Elizabeth M. Keating, Francis Sakita, Ismail Amiri, Getrude Nkini, Melissa H. Watt, Catherine A. Staton, Blandina T. Mmbaga.

**Data curation:** Elizabeth M. Keating, Ismail Amiri, Getrude Nkini, Bryan Young, Blandina T. Mmbaga.

**Formal analysis:** Elizabeth M. Keating, Kajsa Vlasic, Ismail Amiri, Getrude Nkini, Bryan Young.

**Funding acquisition:** Elizabeth M. Keating, Melissa H. Watt, Catherine A. Staton, Blandina T. Mmbaga.

**Investigation:** Elizabeth M. Keating, Kajsa Vlasic, Ismail Amiri, Getrude Nkini, Mugisha Nkoronko, Bryan Young, Jenna Birchall, Melissa H. Watt, Catherine A. Staton.

**Methodology:** Elizabeth M. Keating, Jenna Birchall.

**Project administration:** Elizabeth M. Keating, Francis Sakita, Ismail Amiri, Getrude Nkini, Blandina T. Mmbaga.

**Resources:** Elizabeth M. Keating, Francis Sakita, Ismail Amiri, Getrude Nkini, Melissa H. Watt, Catherine A. Staton, Blandina T. Mmbaga.

**Software:** Elizabeth M. Keating.

**Supervision:** Francis Sakita, Melissa H. Watt, Catherine A. Staton, Blandina T. Mmbaga.

**Visualization:** Elizabeth M. Keating, Kajsa Vlasic, Mugisha Nkoronko, Jenna Birchall.

**Writing – original draft:** Elizabeth M. Keating, Kajsa Vlasic, Jenna Birchall.

**Writing – review & editing:** Francis Sakita, Ismail Amiri, Getrude Nkini, Mugisha Nkoronko, Bryan Young, Melissa H. Watt, Catherine A. Staton, Blandina T. Mmbaga.

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
