## [Decision Letter · Decision Letter 0]

1 Sep 2023

PGPH-D-23-01023

HEALTHCARE PROVIDER PERSPECTIVE ON BARRIERS AND FACILITATORS IN THE CARE OF PEDIATRIC INJURY PATIENTS AT A TERTIARY HOSPITAL IN NORTHERN TANZANIA

Dear Dr. Keating,

Thank you for submitting your manuscript to PLOS Global Public Health. After careful consideration, we feel that it has merit but does not fully meet PLOS Global Public Health’s publication criteria as it currently stands. Therefore, we invite you to submit a revised version of the manuscript that addresses the points raised during the review process.

We look forward to receiving your revised manuscript.

Kind regards,

Hassan Haghparast Bidgoli

Academic Editor

Journal Requirements:

2. Please include a complete copy of PLOS’ questionnaire on inclusivity in global research in your revised manuscript. Our policy for research in this area aims to improve transparency in the reporting of research performed outside of researchers’ own country or community. The policy applies to researchers who have travelled to a different country to conduct research, research with Indigenous populations or their lands, and research on cultural artefacts. The questionnaire can also be requested at the journal’s discretion for any other submissions, even if these conditions are not met.  Please find more information on the policy and a link to download a blank copy of the questionnaire here: https://journals.plos.org/globalpublichealth/s/best-practices-in-research-reporting. Please upload a completed version of your questionnaire as Supporting Information when you resubmit your manuscript.

3. We do not publish any copyright or trademark symbols that usually accompany proprietary names, eg  ©, ®, ™  (e.g. next to drug or reagent names). Please remove all instances of trademark/copyright symbols throughout the text, including © on page 22.

4. We have noticed that you have a list of Supporting Information legends in your manuscript. However, there are no corresponding files uploaded to the submission. Please upload them as separate files with the item type 'Supporting Information'. 

Additional Editor Comments:

Please address the comments raised by the reviewers. In particular,

- Upload a completed SRQR checklist for the reviewers and citing that in the methods

- Provide more details on the composition of the Focus Groups and the participants (e.g., number of each profession, years experience)

- Expand the implications of the findings for the study setting and the context of Tanzania

Reviewers' comments:

Reviewer's Responses to Questions

**Comments to the Author**

1. Does this manuscript meet PLOS Global Public Health’s publication criteria? Is the manuscript technically sound, and do the data support the conclusions? The manuscript must describe methodologically and ethically rigorous research with conclusions that are appropriately drawn based on the data presented.

Reviewer #1: Yes

Reviewer #2: Yes

2. Has the statistical analysis been performed appropriately and rigorously?

Reviewer #1: Yes

Reviewer #2: N/A

3. Have the authors made all data underlying the findings in their manuscript fully available (please refer to the Data Availability Statement at the start of the manuscript PDF file)?

Reviewer #1: Yes

Reviewer #2: No

4. Is the manuscript presented in an intelligible fashion and written in standard English?

Reviewer #1: Yes

Reviewer #2: Yes

5. Review Comments to the Author

Reviewer #1: Thank you for this well written piece of Fogarty sponsored work. Paediatric surgical care in LMICs is significantly behind the care of adult surgical counterparts. Injury has also been a neglected pandemic, and thus, the authors’ work at Kilimonjaro Christian Medical Center is relevant and timely. The choice of qualitative approach is important as in-depth qualitative perspectives to barriers and facilitators to pediatric trauma care in Tanzania have not been previously explored. The rigor of the described methods and the resultant planned publication is commendable. This information might reflect the experience of several other sites in similar settings.

A few minor points to note

1. The authors need to include a checklist (Standards for Reporting Qualitative Research (SRQR)* checklist is preferred- O'Brien BC, Harris IB, Beckman TJ, Reed DA, Cook DA. Standards for reporting qualitative research: a synthesis of recommendations. Academic Medicine, Vol. 89, No. 9 / Sept 2014 DOI: 10.1097/ACM.0000000000000388) as a supporting document to benchmark their work with standards and to guide their write up. The checklist used should be mentioned in their methods.

Consider editing the topic by Standards for Reporting Qualitative Research guidelines- Identifying the study as qualitative or indicating the approach (e.g., ethnography, grounded theory) or data collection methods (e.g., interview, focus group) is recommended.

Introduction- Problem formulation and Purpose or research question are well articulated.

Please clarify (Page 4, Lines 86/87)- *Due to?* the system-level challenges faced by healthcare providers in LMICs, it is important to investigate how these barriers impede quality of care for injured children. It may be best as 2 separate sentences, or even dropping the first half altogether as this has already been established in the introduction.

This study and all procedures were said to have received ethical approval from 2 review bodies- kindly include all ethical approval numbers in the text of the manuscript for clarity.

The research team and reflexivity section appears incomplete as it only focuses on the data collectors. The authorship research team comprises 9 individuals. Consider including a deeper statement on reflexivity including the positionality of the authors/researchers including gender balance as they also interacted with the data in a variety of ways. This will further help us understand the perspectives and biases of the authors with regards to this topic.

Please insert a line in the research methods about the research paradigm (e.g., postpositivist, constructivist/ interpretivist)

Results: Please describe the composition of the Focus Groups in detail. X (y%) nurses, Z (q%) paediatric surgeons, A (b%) GPs etc. This background will help us interpret the findings appropriately.

Table 1 can be brought up- just below the paragraph in which it was first mentioned.

Kindly tell us in more detail what type of practitioners responded in each quote, as these were multidisciplinary Focus Group Discussions. EMD is not detailed enough. Were these responses from the same person? Nurse 1 EMD, Doctor 1 EMD, Doctor 2 EMD etc would add sufficient detail. Same with “Paediatric Ward” and “Surgical Ward” as it would be important to know the training of these practitioners. Health practitioners were defined in the methods as physicians and nurses. The word healthcare providers is nebulous. Defining who is who is very important as perspectives of these groups can be significantly different in some working environments.

Line 266-269 Pediatric-specific subspecialist care, such as pediatric neurosurgeons, pediatric orthopedic surgeons, and pediatric surgeons, are not available at KCMC. The neurosurgeons, orthopedic surgeons, and general surgeons that are able to care for these patients are often very busy given the large volume of cases in both adults and pediatrics. This statement can be strengthened by a quote. The quotation given below focuses on pediatricians.

The final theme seems to be an amalgamation of themes, and the authors should consider the clarity of the codes- “Complexity of cases due to pre-hospital delays in patients presenting for care caused by cultural and financial barriers” focuses on pre-hospital delays (Level 2 specifically- reaching care). However, the discussion of this point centers around level 3 delays (i.e., delays in receiving care) due to financial reasons AFTER the patient has arrived at the hospital and is on the ward or in the operating room. These are distinct concepts and the codes or theme should be reconciled.

Page 19, line 402- widely developed would be preferred to widely adapted, as it comes across as if the authors are suggesting that these guidelines cannot be developed in LMIC settings for LMIC issues, and with LMIC expertise. Line 403 shows that this is not the case.

It is interesting that inter-professional rivalry did not arise as a theme or an issue affecting patient care negatively in these multidisciplinary focus groups as is seen in much of sub-Saharan Africa and beyond. Can the authors comment on that?

Just a note of care- while it is true that learnings from KCMC can possibly be translated to other LMIC settings, the authors have consistently made a jump in the discussion to other LMICs, from the experience of a single Tertiary center in Tanzania. The authors do not deeply discuss the implications for KCMC itself, which is a potentially useful tool for implementation science. As the authors have had dissemination meetings within the hospital, can you please share some of your recommendations (possibly as an additional table, based on the barriers and facilitators identified from this qualitative work). The hospital surgical system and other East African locations can learn from your recommendations.

If nurses and doctors were not balanced in the focus group, then this should be included in the limitations. However, based on thematic saturation that was achieved, the authors do not justify the perceived limitation of a 30 person focus group.

This is a good work and should be published.

Reviewer #2: The authors report a series of focus groups of medical providers involved in pediatric injury care in a tertiary care center in Tanzania in order to identify barriers to the provision of care along with facilitators to the provision of care. They used qualitative analysis methods to identify themes: lack of guidelines, lack of equipment, inadequate staff, and prehospital delays that were felt to impede the proper provision of care to injured children.

The study was well conducted with trained personnel conducting the groups in native language and the subsequent analysis of responses was conducted without identification of specific respondent.

Some questions:

1. Were there opportunities for participants to provide anonymous responses, or responses not while in the presence of other providers, potentially most important for the nurses?

2. What pre-hospital evaluation and transport systems are available? Are there non-cultural/financial burdens to prompt, knowledgeable prehospital care? In HIC more than half of children who die from injury do not survive to reach a hospital. Is the percentage of children who die from an injury prior to arrival to a hospital in Tanzania known?

3. The study deals with care in a tertiary care center. Is that reflective of care more broadly in Tanzania? Could the authors please address that.

4. The recommended various pediatric specialists are lacking from many hospitals in the United States and seems to be very ambitious and potentially unrealistic for Tanzania. Are there other, more realistic options for LMIC? For example, very few pediatric trauma patients require a general surgical operation and could properly trained pediatricians provide sufficient care?

5. Are there pediatric surgeons at KCMC? If so, are they involved in the care of injured children? If not, is the level of injury care demonstrative of the level of overall pediatric care?

6. PLOS authors have the option to publish the peer review history of their article (what does this mean?). If published, this will include your full peer review and any attached files.

**Do you want your identity to be public for this peer review?** For information about this choice, including consent withdrawal, please see our Privacy Policy.

Reviewer #1: No

Reviewer #2: **Yes: **David P. Mooney

---

## [Decision Letter · Decision Letter 1]

17 Oct 2023

PGPH-D-23-01023R1

HEALTHCARE PROVIDER PERSPECTIVE ON BARRIERS AND FACILITATORS IN THE CARE OF PEDIATRIC INJURY PATIENTS AT A TERTIARY HOSPITAL IN NORTHERN TANZANIA: A QUALITATIVE STUDY

Dear Dr. Keating,

Thank you for submitting your manuscript to PLOS Global Public Health. After careful consideration, we feel that it has merit but does not fully meet PLOS Global Public Health’s publication criteria as it currently stands. Therefore, we invite you to submit a revised version of the manuscript that addresses the points raised during the review process.

Please submit your revised manuscript by . If you will need more time than this to complete your revisions, please reply to this message or contact the journal office at globalpubhealth@plos.org. Please include the following items when submitting your revised manuscript:

We look forward to receiving your revised manuscript.

Kind regards,

Hassan Haghparast Bidgoli

Academic Editor

Journal Requirements:

Additional Editor Comments:

Thank you for addressing the comments from the reviewers. Reviewer 1 raised the following minor comments and I would appreciate if you address that and re-submit your manuscript.

In response [to second reviewer] the text on page 19, lines 429-433 was edited to read “There are rarely providers in LMICs with pediatric-specific training in these specialties, '****and expecting this to happen is ambitious and potentially unrealistic for Tanzania.****'" While there is truth in difficulty of meeting this target, the terms "ambitious and potentially unrealistic" used here are strong, and can be viewed as condescending in its presentation. ... Can you kindly advise that it is removed and replaced with less strong language?

Reviewers' comments:

Reviewer's Responses to Questions

**Comments to the Author**

1. If the authors have adequately addressed your comments raised in a previous round of review and you feel that this manuscript is now acceptable for publication, you may indicate that here to bypass the “Comments to the Author” section, enter your conflict of interest statement in the “Confidential to Editor” section, and submit your "Accept" recommendation.

Reviewer #1: All comments have been addressed

2. Does this manuscript meet PLOS Global Public Health’s publication criteria? Is the manuscript technically sound, and do the data support the conclusions? The manuscript must describe methodologically and ethically rigorous research with conclusions that are appropriately drawn based on the data presented.

Reviewer #1: Yes

3. Has the statistical analysis been performed appropriately and rigorously?

Reviewer #1: Yes

4. Have the authors made all data underlying the findings in their manuscript fully available (please refer to the Data Availability Statement at the start of the manuscript PDF file)?

Reviewer #1: Yes

5. Is the manuscript presented in an intelligible fashion and written in standard English?

Reviewer #1: Yes

6. Review Comments to the Author

Reviewer #1: All comments have been adequately addressed or limitations to addressing them clearly stated. Thank you for this.

7. PLOS authors have the option to publish the peer review history of their article (what does this mean?). If published, this will include your full peer review and any attached files.

**Do you want your identity to be public for this peer review?** For information about this choice, including consent withdrawal, please see our Privacy Policy.

Reviewer #1: No

---

## [Editor Report · Decision Letter 2]

23 Oct 2023

HEALTHCARE PROVIDER PERSPECTIVE ON BARRIERS AND FACILITATORS IN THE CARE OF PEDIATRIC INJURY PATIENTS AT A TERTIARY HOSPITAL IN NORTHERN TANZANIA: A QUALITATIVE STUDY

PGPH-D-23-01023R2

Dear Dr. Keating,

We are pleased to inform you that your manuscript 'HEALTHCARE PROVIDER PERSPECTIVE ON BARRIERS AND FACILITATORS IN THE CARE OF PEDIATRIC INJURY PATIENTS AT A TERTIARY HOSPITAL IN NORTHERN TANZANIA: A QUALITATIVE STUDY' has been provisionally accepted for publication in PLOS Global Public Health.

Best regards,

Hassan Haghparast Bidgoli

Academic Editor
